# You Don't Protect if You Don't Expect:
# Breaking the Key Assumption behind CLIP's Test Time Defenses

**Ruize Zhang**[1 2] **Yu Li**[1 2] **Zhang Wan**[1 2] **Juan Cao**[1 2] **Jie Zhang**[1 2] **Sheng Tang**[1 2]

## Abstract

Recent test-time defenses for CLIP claim to preserve zero-shot clean accuracy while improving adversarial robustness. However, we find the reported robustness of six recent proposed state-of-the-art methods substantially overestimated: they fail under basic adaptive attacks. We further observe that these defenses share a common reliance on an *indicative measurement* that is assumed to capture the distributional difference between clean and adversarial samples and to determine whether the defense should preserve or alter the static model's prediction. We argue that this assumption is the fundamental weakness, and we propose CLIP-MAD (Manipulating Assumed Difference), an adaptive attack strategy designed to break it. CLIP-MAD efficiently expands the adversarial distribution without costly full gradient calculations and can be flexibly combined with existing attack baselines to further boost attack strength. Experiments across 13 datasets demonstrate that CLIP-MAD produces strong adversarial samples that markedly reduce the robustness of diverse test-time defenses, revealing a false sense of security in CLIP's zero-shot robustness. Code will be available at https://github.com/rzzhang222/CLIP-MAD.

## 1. Introduction

Large-scale pre-trained vision-language models (VLMs) have attracted considerable research attention (Jia et al., 2021; Ramesh et al., 2021; Li et al., 2022). Among them, CLIP (Radford et al., 2021) stands out for its capacity to perform zero-shot inference by aligning image features with textual prompts. Despite its widespread adoption as a foun-

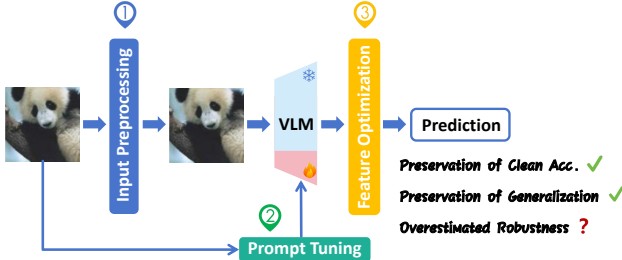

*Figure 1.* Three categories of test-time defenses for CLIP's zero-shot inference. The first category purifies the input before it is fed to the model. The second category adapts the model prompt to fit the test data. The third category optimizes the feature to remove adversarial traits. These test-time defenses achieve impressive performance, but are we overconfident about their robustness?

dation model in many security-sensitive applications, CLIP has demonstrated notable vulnerability to adversarial attacks (Shayegani et al., 2024; Zhao et al., 2023). Consequently, improving the adversarial robustness of CLIP's zero-shot classification (zero-shot robustness) has become a critical problem that has been studied in many previous works (Li et al., 2024; Zhang et al., 2024a; 2025; Mao et al., 2023).

Initial efforts (Li et al., 2024; Zhang et al., 2024a; Mao et al., 2023; Schlarmann et al., 2024; Wang et al., 2024) for CLIP's robustness have adopted the adversarial training (AT) paradigm through adversarial fine-tuning or prompt tuning. Although AT is highly effective in traditional single-modal classification, its application to CLIP introduces significant drawbacks (Xing et al., 2025), most notably overfitting to the fine-tuning dataset and the erosion of pre-trained knowledge.

To circumvent these issues, recent works have shifted to a new paradigm: test-time defenses (Wang et al., 2025; Sheng et al., 2025; Xing et al., 2025; Jiang et al., 2026; Zhang et al., 2025). These methods improve adversarial robustness without significantly degrading clean accuracy or the model's general capabilities, and they are more efficient than traditional test-time defenses (Nie et al., 2022; Zhang et al., 2024b), which often require hundreds or even thousands of times more computation during inference.

The growing prominence of test-time defenses for zero-shot robustness raises a critical question: is their claimed adversarial resilience overestimated? In traditional classification,

[1]Institute of Computing Technology, Chinese Academy of Sciences, Beijing, China [2]University of Chinese Academy of Sciences, Beijing, China. Correspondence to: Yu Li <liyu@ict.ac.cn>.

*Proceedings of the 43rd International Conference on Machine Learning*, Seoul, South Korea. PMLR 306, 2026. Copyright 2026 by the author(s).

previous works (Crose et al., 2022; Carlini et al., 2019; Athalye et al., 2018; Tramer et al., 2020) have established that defenses must withstand defense-specific adaptive attacks to be considered effective, and those that fail are of limited utility. However, to our knowledge, no systematic study has applied this rigorous adversarial scrutiny to test-time defenses in the context of zero-shot inference.

To explore this question, we first take a closer look at six recently proposed state-of-the-art test-time adversarial defenses for CLIP (Wang et al., 2025; Sheng et al., 2025; Xing et al., 2025; Jiang et al., 2026; Zhang et al., 2025) spanning the defense paradigms of input purification, feature space purification, and test-time prompt tuning (shown in Figure 1). A review of the original papers reveals a significant evaluation gap: among these works, TTC (Xing et al., 2025), R-TPT (Sheng et al., 2025), and DOC (Jiang et al., 2026) provide no evaluation against adaptive attacks, while CLIPure-Cos (Zhang et al., 2025), CLIPure-Diff (Zhang et al., 2025), and TAPT (Wang et al., 2025) only report results against ineffective ones. In this paper, we demonstrate that the robust accuracy (classification accuracy on adversarial samples) of most current test-time defenses can be dramatically reduced by straightforward adaptive attacks (i.e., full-gradient and approximated-gradient methods). This finding exposes a critical gap in existing robustness evaluations. To address this inconsistency and enable more reliable assessment, we develop a flexible adaptive attack strategy applicable across diverse CLIP defenses. By analyzing the common principles underlying these methods, we identify a core vulnerability: their reliance on a fundamental indicative-measurement assumption. Then we propose the adaptive attack strategy CLIP-MAD (Manipulating Assumed Difference), which directly targets this inherent flaw.

The advantages of our CLIP-MAD are two-folds. First, it targets a fundamental component in the pipeline of CLIP's test-time defenses rather than a specific defense technique, which grants it inherent adaptability to future defense variants. Despite the defenses' technical diversity spanning input purification, feature optimization, and prompt tuning, this indicative measurement guides subsequent optimization steps toward preserving clean features or modifying adversarial ones, thereby approximating the ideal of correcting adversarial predictions while leaving clean ones intact. In the setting of zero-shot inference with CLIP, however, formulating this measurement requires prior assumptions about adversarial characteristics. We identify this as a core vulnerability: when the assumed measurement deviates from the true adversarial distribution, it induces erroneous behavior in the defense. Our CLIP-MAD attacks this Achilles' heel.

Second, our CLIP-MAD attack is efficient and effective. While future test-time defenses for CLIP may arbitrarily combine non-differentiable or computationally heavy iterative components to evade basic full-gradient adaptive attacks, our method targets the key assumption before the actual optimization of the defense and provides a way to construct attacks while bypassing such components. Even when a defense is differentiable or approximately differentiable, it may contain components that are burdensome and detrimental to attacks but essential for the defense (Crose et al., 2022). A full-gradient or gradient-approximation attack can thus be trapped in a local optimum and require much more computation than the defender, limiting its effectiveness. Our CLIP-MAD not only mitigates this issue by finding a less affected adversarial direction, but also avoids the need to design appropriate substitutes for arbitrary defensive operations and to run extra attack iterations in backward approximation method (Athalye et al., 2018). Furthermore, by exploiting the defense's core assumption, our method can craft adversaries that induce unexpected failure modes, effectively enriching the adversarial distribution beyond the defense's intended design.

We emphasize that our goal is not to invalidate existing defenses but to build upon their insights in pursuit of more reliable zero-shot robustness. Accordingly, this work contributes a methodological tool for evaluating worst-case adversarial resilience, enabling more rigorous assessments of current and future test-time defenses. Finally, we derive practical insights and recommendations for both designing robust defenses and conducting effective adaptive evaluations in the zero-shot inference paradigm.

The main contributions of this paper are:

- We identify and analyze a core vulnerability in the general pipeline of CLIP's test-time defenses, which stems from their reliance on the fundamental indicative-measurement assumption.

- We propose a novel adaptive attack named CLIP-MAD. Designed to be computationally efficient and easily adaptable, it can be flexibly combined with existing attack methods to significantly enhance adversarial strength, making it a valuable tool for rigorous, worst-case robustness evaluation.

- Experimental results demonstrate the efficacy of our method in improving attack strength or efficiency against CLIP's test-time defenses across 13 datasets.

## 2. Related Works

**Train-Time Adversarial Defenses of CLIP** Some researchers (Li et al., 2024; Zhang et al., 2024a; Mao et al., 2023; Schlarmann et al., 2024; Wang et al., 2024) have transformed adversarial training into adversarial fine-tuning for pretrained VLM models. TeCoA (Mao et al., 2023) proposes to use prompted texts for adversarial data generation.

PMG-AFT (Wang et al., 2024) adds two additional learning objectives as regularization to keep the generalization capability of the original CLIP. FARE (Schlarmann et al., 2024) proposes an unsupervised way for adversarial sample generation to alleviate the problem of overfitting and clean accuracy degradation. To mitigate the problem of heavy computation in AT, APT (Li et al., 2024) and AdvPT (Zhang et al., 2024a) propose to adversarially tune the prompts only.

**Test-Time Adversarial Defenses of CLIP**    Researchers have found that train-time adversarial tuning may devastate generalization abilities in large-scale pre-trained models, turning to test-time defenses. Among test-time defenses, TTC (Xing et al., 2025) follows the paradigm of input pre-processing and purifies the input image by applying counter-attack perturbation. DOC (Jiang et al., 2026) refines the drawbacks of TTC with more advanced counterattack. R-TPT (Sheng et al., 2025) and TAPT (Wang et al., 2025) follow the paradigm of test-time prompt tuning with different learning objectives. CLIPure-Cos (Zhang et al., 2025) and CLIPure-Diff (Zhang et al., 2025) follow the paradigm of feature purification using likelihood estimation. Test-time defenses have multiple advantages compared to training-time methods as in Figure 1 and we will focus on their robustness. A more detailed review for these methods will be provided later.

**Adaptive Attacks**    Adaptive attacks are the adversarial attacks crafted for specific defense (Crose et al., 2022; Carlini et al., 2019; Athalye et al., 2018; Tramer et al., 2020). Previous research has shown that some test-time defenses in traditional classification can be broken by adaptive attacks, highlighting the importance of understanding the limitations of newly proposed defenses. Carlini & Wagner (2017) finds that ten adversarial detection methods can be bypassed by carefully constructed adversaries. Although no standard evaluation procedure is proposed, Tramer et al. (2020) suggests pursuing simplicity in the attack design and Crose et al. (2022) recommends full-gradient and approximated-gradient attack for defense assessment. These works inspire us to explore the resilience of the newly proposed test-time defenses in zero-shot inference. In this work, we provide a generalizable method that further boosts the adaptive attack strength in CLIP-based zero-shot classification.

## 3. Preliminary

Given an image encoder $f$ and a text encoder $g$, CLIP computes the logits of image $\mathbf{x}$ belonging to a certain class as:

$$\text{logit}_{c_i}(\mathbf{x}) = \cos\left(f(\mathbf{x}), g(c_i)\right), \tag{1}$$

where $c_i$ is the text of the candidate class prompted by a template that can take the form of "a photo of a $[c_i]$:".

In adversarial attacks, a bounded perturbation $\boldsymbol{\delta}$ is added to a clean test sample $\mathbf{x}$ with ground truth label $T(\mathbf{x})$ to generate an adversarial sample $\mathbf{x}_{\text{adv}}$ by:

$$\mathbf{x}_{\text{adv}} = \text{clamp}(\mathbf{x} + \boldsymbol{\delta}_{\text{adv}}, 0, 1), \tag{2}$$

where

$$\boldsymbol{\delta}_{\text{adv}} = \arg\max_{\boldsymbol{\delta}} \mathcal{L}(\text{logit}(\mathbf{x} + \boldsymbol{\delta}), T(\mathbf{x})) \text{ s.t. } |\boldsymbol{\delta}|_p \le \epsilon. \tag{3}$$

To achieve zero-shot robustness, test-time defenses generally span three techniques targeting different steps of classification, which are input preprocessing, test-time prompt tuning, and feature optimization. Given a newly proposed CLIP's test-time defense potentially contain these techniques and test data $\mathbf{x}$, we can generally write the predicted result for class $c_i$ in the format of:

$$\text{logit}_{c_i}(\mathbf{x}) = \cos(\tilde{f}(\tau(\mathbf{x})), g_{\text{tuned}}(c_i)). \tag{4}$$

where $\tau(\mathbf{x})$ denotes the preprocessed input, $\tilde{f}(\cdot)$ denotes the optimized image feature extractor and $g_{\text{tuned}}$ is the text encoder with the tuned prompts.

## 4. Adaptive Attack via Manipulating Assumed Difference

### 4.1. General Idea of CLIP-MAD

Given the variation in defense paradigms and techniques, previous work (Crose et al., 2022) in traditional adversarial learning finds it hard to provide standard instructions for constructing stronger adaptive attacks than the basic ones. However, it is pointed out that basic adaptive attacks alone are incompetent for evaluating test-time defenses' worst-case robustness, demonstrating the difficulty and need for effective adaptive attacks.

In this paper, we focus on designing an adaptive attack strategy for test-time defenses of CLIP's zero-shot classification, which can be easily computed and adapted to different defenses. Figure 2 presents the general idea of our adaptive attack strategy and its difference from full gradient attack. We start by analyzing the general defense pipeline. Given a CLIP's test-time defense, ideally it gives the same output as the static model (CLIP model without defense) when the input is clean and different output from the static model when the input is an adversary:

$$g \cdot f(\mathbf{x}) \begin{cases} = f(\mathbf{x}) & \text{if } \mathbf{x} \text{ is clean,} \\ \neq f(\mathbf{x}) & \text{if } \mathbf{x} \text{ is an adversary,} \end{cases} \tag{5}$$

where $f$ is the original CLIP model, $g$ is the specific test-time defense and $\mathbf{x}$ is the visual input. $f(\mathbf{x})$ stands for the static feature and $g \cdot f(\mathbf{x})$ stands for the protected feature.

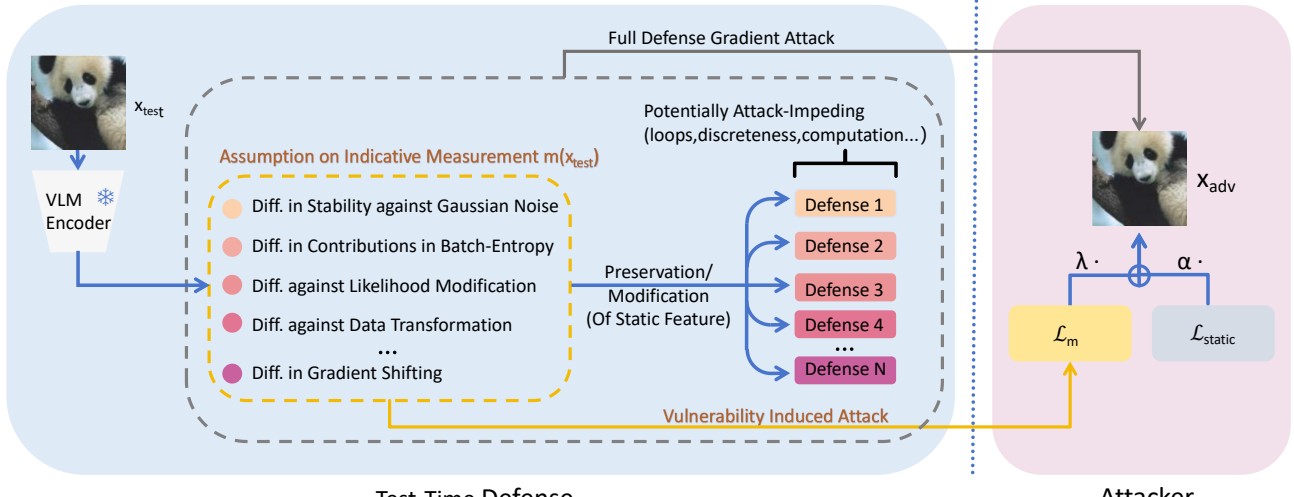

*Figure 2.* An illustration of the idea of our proposed adaptive attack strategy. The left part of the figure shows the general pipeline of CLIP's test-time defenses and the right part presents our CLIP-MAD attack. Our method enhances the flexibility and strength of adaptive attacks via breaking the distributional assumption on the indicative measurement at the core of the general test-time defense pipeline.

In real-world testing scenario, the ground truth label indicating whether a test sample is clean or adversarial is unavailable. Therefore, the defense must decide whether to preserve or modify the static feature base on an indicative trait measured from the test input. To enable CLIP's zero-shot inference across datasets, this step requires prior adversarial knowledge or assumption about the distribution of this indicative trait. Without loss of generalizability, we denote the indicative measurement utilized by the defense to induce this behavior discrepancy as $m(\mathbf{x})$, whose value depends on $\mathbf{x}$. While $\mathbf{x}$ can be either clean or adversarial, there exists the following assumption for $m$ in the defense:

$$m(\mathbf{x}_{\text{clean}}) \sim p_{\phi_{\text{clean}}}, m(\mathbf{x}_{\text{adv}}) \sim q_{\phi_{\text{adv}}}. \quad (6)$$

Then for CLIP's test-time defenses, we typically have:

$$g \cdot f(\mathbf{x}) \begin{cases} = f(\mathbf{x}), & \text{if } p_{\phi_{\text{clean}}}(m(\mathbf{x})) > q_{\phi_{\text{adv}}}(m(\mathbf{x})), \\ \neq f(\mathbf{x}), & \text{otherwise.} \end{cases} \quad (7)$$

When a test input X has "adversarial trait" assumed by the defense, i.e., $p_{\phi_{\text{clean}}}(m(\mathbf{x})) \leq q_{\phi_{\text{adv}}}(m(\mathbf{x}))$, the defense generally maximizes the difference from static features (up to a threshold) to suppress the adversarial effects. Otherwise static feature is preserved. For the defense to be effective, the distributions $p$ and $q$ should be separable:

$$m(\mathbf{x}_{\text{clean}}) \sim p_{\phi_1}, m(\mathbf{x}_{\text{adv}}) \sim q_{\phi_2}$$
$$\text{s.t.} \int \min(p, q) \mathrm{d}(m) \approx 0, \quad (8)$$

where $\min(p, q)$ denotes the minimum of the probability density of $p$ and $q$ on $m$. Then we search for a candidate $\mathbf{x}_{\text{CLIP\_MAD}}$ satisfying the requirement of deviating

$m(\mathbf{x}_{\text{CLIP\_MAD}})$ from its assumed distribution:

$$\boldsymbol{\delta}_{\text{CLIP\_MAD}} = \text{argmax}_{\boldsymbol{\delta}} |m(\mathbf{x} + \boldsymbol{\delta}) - m(\mathbf{x})|, \quad (9)$$

$$\mathbf{x}_{\text{CLIP\_MAD}} = \mathbf{x} + \boldsymbol{\delta}_{\text{CLIP\_MAD}}. \quad (10)$$

This unexpected measurement value serves to create the confusion between clean and adversarial features, i.e.:

$$p_{\phi_{\text{clean}}}(m(\mathbf{x}_{\text{adv}} + \boldsymbol{\delta}_{\text{CLIP\_MAD}})) > p_{\phi_{\text{clean}}}(m(\mathbf{x}_{\text{adv}})), \quad (11)$$

$$q_{\phi_{\text{adv}}}(m(\mathbf{x}_{\text{clean}} + \boldsymbol{\delta}_{\text{CLIP\_MAD}})) > q_{\phi_{\text{adv}}}(m(\mathbf{x}_{\text{clean}})). \quad (12)$$

This in turn leads to undesirable preserving and modifying behaviors in the defense, making more adversarial feature preserved and more clean feature removed:

$$|g \cdot f(\mathbf{x}_{\text{adv}} + \boldsymbol{\delta}_{\text{CLIP\_MAD}}) - f(\mathbf{x}_{\text{adv}} + \boldsymbol{\delta}_{\text{CLIP\_MAD}})|$$
$$< |g \cdot f(\mathbf{x}_{\text{adv}}) - f(\mathbf{x}_{\text{adv}})| \approx max|\Delta f(\mathbf{x})| \quad (13)$$

and

$$|g \cdot f(\mathbf{x}_{\text{clean}} + \boldsymbol{\delta}_{\text{CLIP\_MAD}}) - f(\mathbf{x}_{\text{clean}} + \boldsymbol{\delta}_{\text{CLIP\_MAD}})|$$
$$> |g \cdot f(\mathbf{x}_{\text{clean}}) - f(\mathbf{x}_{\text{clean}})| \approx 0, \quad (14)$$

where $|\Delta f(\mathbf{x})|$ stands for the variation from static feature.

To find such an assumption-breaking $\mathbf{x}_{\text{CLIP\_MAD}}$ that possesses adversarial effects, we make a modification of the adversarial data generation objective from solely maximizing the classification loss to jointly maximizing the sum of two compositional objectives: one basic static classification loss to ensure adversarial effects and one $m(\mathbf{x})$ adjusting loss to break the assumed distributional difference. The general formulation for CLIP-MAD is:

$$\mathcal{L}_{\text{CLIP\_MAD}} = \alpha \cdot \mathcal{L}_{\text{static}} + \lambda \cdot \mathcal{L}_m, \quad (15)$$

where $\mathcal{L}_{\text{static}}$ is the static model classification loss and $\mathcal{L}_m$ is or has an equivalent effect of $|m(\mathbf{x} + \delta) - m(\mathbf{x})|$. Hyperparameter $\alpha$ decides whether to include the static loss and $\lambda$ balances the two terms if $\alpha$ is not 0. $\alpha$ and $\lambda$ are easy to set without complex tuning but can change across the defenses.

In the following sections, we will use case studies to illustrate how to apply our CLIP-MAD to State-of-the-Art defenses with simple adaptation of $\mathcal{L}_m$. For each test-time defense, we first give a brief summary, then we provide the attacking results with CLIP-MAD.

## 4.2. Case Studies of SOTA Test-time Defenses

### 4.2.1. ANALYSIS AND ATTACK OF TTC

**Summary of Defense**    TTC (Xing et al., 2025) is built on the observation that adversarial features are trapped in a falsely stable area less sensitive to tiny random noise. The ratio r of $L_2$ feature difference caused by tiny noise is defined as:

$$r(\mathbf{x}) = (|f(\mathbf{x} + n)| - |f(\mathbf{x})|)/|f(\mathbf{x})|, \qquad (16)$$

where $f$ is the visual encoder and n is a random noise. If difference ratio $r$ is less than a threshold, the static feature undergoes a purification process that exerts counter-attack perturbation with largest distortion in a self-supervised way:

$$\boldsymbol{\delta}_{ttc} = \text{argmax}_{\boldsymbol{\delta}} |f(\mathbf{x} + \boldsymbol{\delta}) - f(\mathbf{x})| \text{ s.t. } |\boldsymbol{\delta}|_p \leq \epsilon. \quad (17)$$

The original paper of TTC has proposed a full-gradient attack considering the defense. However, no result against this adaptive attack is provided.

**Attack with CLIP-MAD**    It's straightforward to see the difference ratio r utilized by TTC constitutes the key measurement here, i.e., $m(\mathbf{x}) = r$. The key of attacking the defense is to devastate this difference assumption. The most natural objective of deviating this $m(\mathbf{x})$ while keep the original prediction is:

$$\mathcal{L}_m = r(\mathbf{x}_{\text{CLIP\_MAD}}) + \cos(f(\mathbf{x}_{\text{CLIP\_MAD}}), f(\mathbf{x})). \quad (18)$$

In experiment, we notice that we don't even need this explicit $r$ term for difference ratio minimization. An even simpler objective that exerts a perturbation but just keep the perturbed feature close to the original clean one is enough to implicitly make $m(\mathbf{x})$ less than 0.2, which is the threshold set by TTC. Thus we directly use this minimum modification objective to the original image for $\mathcal{L}_m$:

$$\mathcal{L}_m = \cos(f(\mathbf{x}_{\text{CLIP\_MAD}}), f(\mathbf{x})). \quad (19)$$

Since this loss term already bypasses the adversarial identification mechanism of TTC, extra static classification loss is not included. Thus we set $\alpha$ to 0 and $\lambda$ to 1.

Note that our attack objective finds a different distribution of adaptive adversaries that fools the defense into feature modification but is close to original clean data. Thus in this case we apply our CLIP-MAD as a complement of full-gradient attack and the results are denoted as CLIP-MAD in Table 1. In comparison, the results of static adversarial attack and full-gradient attack are also reported as Static and Full. Given the stochasticity introduced by counter-attack noise, the full gradient attack is already integrated with Expectation over Transformation (EOT-5). Our CLIP-MAD further reduces robust accuracy from 11.0% against full-gradient attack to 4.7% in average. Note that most clean samples are falsely identified as adversaries in just one attacking iteration and we skip all following iterations and defense gradient calculation in CLIP-MAD, making this added calculation much smaller than that required by full-gradient adaptive attack.

### 4.2.2. ANALYSIS AND ATTACK OF DOC

**Summary of Defense**    DOC (Jiang et al., 2026) is a state-of-the-art CLIP's test-time defense that exerts conter-attack perturbation like TTC does but with more advanced optimization techniques. To avoid overfitting of counter-attack direction to the local feature deviation loss, DOC proposes to use the combination of momuntum-update strategy and random perturbation direction orthorgonal to the primary gradient to generate diverse and strong counter-attack perturbations. Furthermore, to prevent the potential vulnerability against adversarial samples with low difference ratio, the single threshold is replaced by a sigmoid soft-gating weight that dynamically adjusts the proportion of counter-attack perturbation:

$$w = \sigma(\gamma \cdot (\tau - r(x))) \in (0, 1), \qquad (20)$$

where $\tau$ is the threshold, $\gamma$ controls sharpness and $\sigma$ is the sigmoid function. The final counterattack perturbation is:

$$\boldsymbol{\delta}_{doc} = w \cdot \boldsymbol{\delta}_{ca} + (1 - w) \cdot \boldsymbol{\delta}_0, \qquad (21)$$

where $\boldsymbol{\delta}_{ca}$ is the diversified counterattack perturbation that maximizes the feature difference and $\boldsymbol{\delta}_0$ is random noise. The adversarial robustness reported is much higher than that of TTC, showing state-of-the-art performance.

**Attack with CLIP-MAD**    Despite the improved gradient update techniques in DOC, we focus on the exploited distributional difference on measurement $m(\mathbf{x})$. Although the hard threshold for identifying adversarial data is replaced by a soft-gating technique, the original implementation of DOC still uses the feature difference against random noise as $m(\mathbf{x})$ to measure the amount of adversarial trait and strength of counterattack needed. Thus we once again use the objective we developed for TTC to induce false modification in clean features, but use all the attack iterations to

*Table 1.* Zero-shot robust accuracies under PGD-10 attack with $\epsilon$=4.0/255 for defense method TTC.

| Attack | CIFAR10 | CIFAR100 | STL10 | Food101 | Oxford | Flowers102 | EuroSAT | TinyImageNet | ImageNet | Caltech101 | Caltech256 | StanfordCars | PCAM | Avg |
|---|---|---|---|---|---|---|---|---|---|---|---|---|---|---|
| Static | 31.2 | 12.3 | 52.0 | 16.4 | 21.3 | 14.6 | 5.6 | 5.2 | 23.1 | 56.2 | 34.0 | 9.2 | 52.4 | 25.7 |
| Full | 12.8 | 2.7 | 24.3 | 4.4 | 7.9 | 5.8 | 3.6 | 0.8 | 10.4 | 24.2 | 12.1 | 3.9 | 30.0 | 11.0 |
| **CLIP-MAD (ours)** | **3.6** | **0.6** | **9.4** | **0.0** | **1.1** | **1.4** | **0.4** | **0.0** | **1.9** | **13.6** | **5.3** | **0.1** | **23.2** | **4.7** |

*Table 2.* Zero-shot robust accuracies under PGD-10 attack with $\epsilon$=4.0/255 for defense method DOC.

| Attack | CIFAR10 | CIFAR100 | STL10 | Food101 | Oxford | Flowers102 | EuroSAT | TinyImageNet | ImageNet | Caltech101 | Caltech256 | StanfordCars | PCAM | Avg |
|---|---|---|---|---|---|---|---|---|---|---|---|---|---|---|
| Static | 36.1 | 14.3 | 68.8 | 34.5 | 42.5 | 28.3 | 5.9 | 8.5 | 34.9 | 76.8 | 45.3 | 20.0 | 57.1 | 36.4 |
| Full | 17.6 | 7.9 | 43.6 | 14.5 | 16.5 | 11.9 | 2.3 | 2.7 | 18.5 | 52.5 | 26.2 | 7.3 | 32.6 | 19.5 |
| **CLIP-MAD (ours)** | **6.4** | **1.8** | **24.4** | **1.1** | **5.8** | **3.8** | **0.4** | **0.2** | **5.6** | **34.2** | **11.5** | **0.9** | **26.0** | **9.4** |

make $m(\mathbf{x})$ even lower to break the soft gating mechanism. We use the same $\mathcal{L}_m$, $\alpha$ and $\lambda$ as that for TTC in last section:

$$\mathcal{L}_m = \cos(f(\mathbf{x}_{\text{CLIP\_MAD}}), f(\mathbf{x})). \qquad (22)$$

The sophisticated counterattack techniques in DOC are oppositely utilized by our attack to falsely push away clean features and our CLIP-MAD is useful even without full details of this test-time optimization. Similar to the case of TTC, full gradient attack is already integrated with EOT-5. We use CLIP-MAD as a complement of full-gradient attack and the results are denoted as CLIP-MAD in Table 2. CLIP-MAD further reduces the robust accuracy from 19.5% against full-gradient attack to 9.4% in average.

### 4.2.3. ANALYSIS AND ATTACK OF R-TPT

**Summary of Defense**  R-TPT (Sheng et al., 2025) is a test-time defense method for CLIP that follows the paradigm of test-time prompt tuning. The method is built on the observation that adversarial samples vary significantly in their preserved adversarial traits after complex data augmentations. Thus it applies AugMix (Hendrycks et al., 2020) to the input 64 times. The augmented views with largest prediction confidence are selected to tune the model prompt with the pointwise entropy minimization objective:

$$\mathcal{L}_{\text{pointwise}} = \frac{1}{|\text{B}|} \sum_{b=1}^{\text{B}} \mathcal{H}(p^b), \qquad (23)$$

where $\mathcal{H}$ denotes the Shannon entropy and $|\text{B}|$ is the augmentation batch size. Since the adversarial views generally has smaller entropy, their effects in the entropy-minimizing prompt tuning process are dominated by augmented samples with reduced adversarial trait. After the prompt tuning, the predictions of augmented views are selected and weighted based on the similarities between closest neighbor pairs to filter out outlier predictions.

**Attack with CLIP-MAD**  R-TPT assumes two distributional difference between clean and adversarial data: 1) small v.s. large prediction shift before and after complex augmentation and 2) small v.s. large prediction entropy. Here we attack the assumed cross-augmentation prediction shift at the very beginning of the defense to skip all the defense gradient calculation including prompt tuning, index selecting and reweighting.

Note that both the AugMix data augmentations and index selection based weighted prediction in this defense bring difficulties to construct full-defense gradient attack. Our method is especially simple and efficient in such cases. We denote the logit shift after augmentation as $m(\mathbf{x})$. To induce deviation of $m(\mathbf{x})$ during attack, we have:

$$\begin{aligned}
&\text{argmax}|m(\mathbf{x}_{\text{adv}} + \boldsymbol{\delta}_{\text{CLIP\_MAD}}) - m(\mathbf{x}_{\text{adv}})| \\
&= \text{argmin}|m(\mathbf{x}_{\text{adv}} + \boldsymbol{\delta}_{\text{CLIP\_MAD}}) - m(\mathbf{x}_{\text{clean}})| \\
&\approx \text{argmin}|m(\mathbf{x}_{\text{adv}} + \boldsymbol{\delta}_{\text{CLIP\_MAD}})| \\
&= \text{argmin} \sum_{\tau_i} \text{CE}(f(\tau_i(\mathbf{x}_{\text{CLIP\_MAD}})), T_{\text{adv}}) \\
&= \text{argmax} \sum_{\tau_i} \text{CE}(f(\tau_i(\mathbf{x}_{\text{CLIP\_MAD}})), T(\mathbf{x})),
\end{aligned} \qquad (24)$$

where $T_{adv}$ is the predicted adversarial label and $T(\mathbf{x})$ is the ground truth label. Then we have:

$$\mathcal{L}_m = \sum_{\tau_i} \text{CE}(\tau_i(\mathbf{x}_{\text{CLIP\_MAD}}), T(\mathbf{x})). \qquad (25)$$

During the attack, we replace the AugMix transformation with differentiable random resized crop. Note that we don't change or replace anything in the defense and this augmentation replacement only occurs in our attack generation. This $m$-targeting loss restricts the abnormal prediction shift $m(\mathbf{x}_{adv})$ induced by adversarial perturbations, invalidating all following steps in the defense. We assign equal weights for all the views in the batch as this is the most natural way.

*Table 3.* Zero-shot robust accuracies under PGD-10 attack with $\epsilon$=4.0/255 for defense method R-TPT. Note that full defense gradient attack is not provided here due to the non-differentiability.

| Attack | CIFAR10 | CIFAR100 | STL10 | Food101 | Oxford | Flowers102 | EuroSAT | TinyImageNet | ImageNet | Caltech101 | Caltech256 | StanfordCars | PCAM | Avg |
|---|---|---|---|---|---|---|---|---|---|---|---|---|---|---|
| Static | 31.8 | 17.7 | 75.5 | 45.4 | 41.8 | 39.9 | 2.0 | 15.0 | 32.8 | 82.1 | 56.2 | 16.1 | 31.3 | 37.5 |
| **CLIP-MAD (Ours)** | **0.0** | **0.0** | **2.0** | **0.0** | **0.0** | **2.8** | **0.0** | **0.0** | **0.5** | **3.3** | **0.4** | **0.3** | **0.3** | **0.7** |

Thus we set both $\alpha$ and $\lambda$ to 1 naturally. In the experiment, we use our CLIP-MAD as a complement of static attack. The results of static attack is provided for comparison in Table 3. Our adaptive attack skips all discrete operations in the defense and successfully reduce robust accuracy from 37.5% to around zero.

### 4.2.4. ANALYSIS AND ATTACK OF TAPT

**Summary of Defense**   As another test-time prompt tuning based defense, TAPT (Wang et al., 2025) first generates augmented views of input image with random resized crop and horizontal flip, then tunes multi-layer visual and textual tokens by augmented views with lowest entropy. Unlike R-TPT, TAPT is differentiable, the batchwise-entropy calculated with averaged predicted probabilities instead of the pointwise entropy is calculated as the prompt tuning objective:

$$\mathcal{L}_{entropy} = \mathcal{H}(\bar{p}), \quad (26)$$

$$\bar{p} = \frac{1}{|\mathbf{B}|} \sum_{b=1}^{B} p^b, \quad (27)$$

where $\mathcal{H}$ is the Sannon entropy and $|\mathbf{B}|$ is the augmentation batch size. During inference, the predicted logits of adversarial views are outweighted by clean views in the averaged prediction probabilities and the prompt is tuned in the direction of confident prediction of clean views.

The method also applies an alignment of mean and variance statistics with robustly tuned models, thus it can be regarded as a combination of train-time defense and test-time defense. We first trained the adversarial model as described in TAPT and implemented TAPT-VLI, which enables independent multi-layer visual and textual token tuning.

**Attack with CLIP-MAD**   Similar to the case of R-TPT, the tuning direction is only valid if the predicted logits of adversarial views degrade after strong augmentation. Otherwise the averaged entropy minimization misleads the prompt tuning into adversarial direction. Thus the defense is still based on key measurement $m(\mathbf{x})$ of logit shift after augmentation. We once again attack this key assumption and keep the values of $\alpha$ and $\lambda$ the same as in last section, i.e.:

$$\mathcal{L}_m = \sum_{\tau_i} \text{CE}(\tau_i(\mathbf{x}_{\text{CLIP\_MAD}}), T(\mathbf{x})). \quad (28)$$

The experimental results are shown in Table 4. one can notice full gradient attack already reduces the robustness to around zero, showing that the prompt tokens are tuned in adversarial directions despite the initialization and statistics alignment with adversarially trained model. In comparison, CLIP-MAD is able to successfully attack this defense without even visiting the multi-layer prompt tuning operations.

### 4.2.5. ANALYSIS AND ATTACK OF CLIPURE-DIFF AND CLIPURE-COS

**Summary of Defense**   CLIPure-Cos and CLIPure-Diff are two versions of CLIPure (Zhang et al., 2025). These two defenses are based on the observation that adversarial data doesn't reside on the clean data manifold and has a likelihood lower than clean ones. Thus these two defenses purify the feature by optimizing its estimated likelihood. While CLIPure-Diff relies on an extra DifussionPrior network to estimate feature likelihood, CLIPure-Cos uses cosine similarity between the image feature and the blank text template to estimate the likelihood of the feature and achieves unprecedented purification efficiency. The feature $\boldsymbol{z}$ is then updated as:

$$\boldsymbol{u} \leftarrow \boldsymbol{u} + \eta \frac{\partial \log(p(\boldsymbol{z}))}{\partial \boldsymbol{z}} \cdot \frac{\partial \boldsymbol{z}}{\partial \boldsymbol{u}}, \quad (29)$$

where

$$\boldsymbol{u} = \boldsymbol{z}/||\boldsymbol{z}||_2. \quad (30)$$

With this purification operation, CLIPure-Diff and CLIPure-Cos improve robustness while preserving clean accuracy.

**Attack with CLIP-MAD**   Although CLIPure-Diff and CLIPure-Cos achieve impressive robustness against full-defense gradient adaptive attack, we find static model attack, which is assumed to be weaker than adaptive attack, dramatically reduces the robust accuracy. This phenomenon is analyzed in previous work (Athalye et al., 2018) as gradient obfuscation, a common pitfall in traditional test-time defenses. In this cases, the ineffective full-gradient attack implies the domination of gradient updates by large feature oscillations in the defense instead of classification loss.

For these two defenses, clean features are supposed to have consistent predictions before and after this likelihood optimization and adversarial features are supposed to have

*Table 4.* Zero-shot robust accuracies under PGD-10 attack with $\epsilon$=4.0/255 for defense method TAPT-VLI, the multi-layer visual and text independent tuning version of TAPT. Note that our CLIP-MAD attack skips all the operations of multi-layer token tuning.

| Attack | CIFAR10 | CIFAR100 | STL10 | Food101 | Oxford | Flowers102 | EuroSAT | TinyImageNet | ImageNet | Caltech101 | Caltech256 | StanfordCars | PCAM | Avg |
|---|---|---|---|---|---|---|---|---|---|---|---|---|---|---|
| Static | 21.8 | 12.1 | 72.5 | 27.9 | 14.5 | 18.2 | 0.3 | 8.3 | 35.1 | 90.1 | 58.5 | 8.6 | 26.3 | 30.3 |
| Full | 0.0 | 0.0 | **0.0** | 0.0 | 0.0 | 0.0 | 0.0 | 0.0 | **0.0** | 0.0 | **4.3** | 0.0 | 0.0 | **0.3** |
| **Ours** | **0.0** | **0.0** | 0.1 | **0.0** | **0.0** | **0.0** | **0.0** | **0.0** | 0.1 | **0.0** | 5.2 | **0.0** | **0.0** | 0.4 |

*Table 5.* Zero-shot robust accuracies under PGD-10 attack with $\epsilon$=1.0/255 for defense methods CLIPure-Cos and CLIPure-Diff.

| Defense | Attack | CIFAR10 | CIFAR100 | STL10 | Food101 | Oxford | Flowers102 | EuroSAT | TinyImageNet | ImageNet | Caltech101 | Caltech256 | StanfordCars | PCAM | Avg |
|---|---|---|---|---|---|---|---|---|---|---|---|---|---|---|---|
| | Full | 92.1 | 72.4 | 98.6 | 89.5 | 84.1 | 76.1 | 42.5 | 60.1 | 86.4 | 99.5 | 94.8 | 54.7 | 63.6 | 78.0 |
| CLIPure-Cos | Static | 3.0 | 1.0 | 26.0 | 0.4 | 0.5 | 3.4 | 0.0 | 0.7 | 2.5 | 29.0 | 19.0 | 0.0 | 0.0 | 6.6 |
| | **Ours** | **2.3** | **0.4** | **20.5** | **0.0** | **0.0** | **1.8** | **0.0** | **0.1** | **1.6** | **20.6** | **14.7** | **0.0** | **0.0** | **4.8** |
| | Full | 94.4 | 76.1 | 96.1 | 95.3 | 77.1 | 83.5 | 88.6 | 66.7 | 86.9 | 82.5 | 95.6 | 54.9 | 65.5 | 81.8 |
| CLIPure-Diff | Static | 8.2 | 2.3 | 30.9 | 0.9 | **0.6** | 5.5 | 0.4 | 1.8 | 3.9 | 17.7 | 22.0 | 0.1 | 0.5 | 7.3 |
| | **Ours** | **7.1** | **0.9** | **27.5** | **0.4** | 0.8 | **3.5** | **0.0** | **1.3** | **3.1** | **12.8** | **20.9** | **0.0** | **0.0** | **6.0** |

purified predictions. It's straightforward to see the key measurement $m(\mathbf{x})$ is the difference in prediction shift after likelihood modification. We denote $\mathbf{x}$ with one approximated likelihood modification gradient descent step as $\mathbf{x} + \nabla p(\mathbf{x})$. By steps similar to equation (24), we have:

$$
\begin{aligned}
&\mathrm{argmax}|m(\mathbf{x}_{\mathrm{adv}} + \boldsymbol{\delta}_{\mathrm{CLIP\_MAD}}) - m(\mathbf{x}_{\mathrm{adv}})| \\
&= \mathrm{argmin}|m(\mathbf{x}_{\mathrm{adv}} + \boldsymbol{\delta}_{\mathrm{CLIP\_MAD}}) - m(\mathbf{x}_{\mathrm{clean}})| \\
&\approx \mathrm{argminCE}(f(\mathbf{x}_{\mathrm{CLIP\_MAD}} + \nabla p(\mathbf{x}_{\mathrm{CLIP\_MAD}})), T_{\mathrm{adv}}) \\
&= \mathrm{argmaxCE}(f(\mathbf{x}_{\mathrm{CLIP\_MAD}} + \nabla p(\mathbf{x}_{\mathrm{CLIP\_MAD}})), T(\mathbf{x})),
\end{aligned}
\tag{31}
$$

and

$$
\mathcal{L}_m = \mathrm{CE}(f(\mathbf{x}_{\mathrm{CLIP\_MAD}} + \nabla p(\mathbf{x}_{\mathrm{CLIP\_MAD}})), T(\mathbf{x})). \tag{32}
$$

Similar to the case of R-TPT, we naturally set $\alpha$ and $\lambda$ to 1 to give equal emphasis on different views. Table 5 shows the efficacy of our CLIP-MAD even when full-gradient attack fails due to gradient obfuscation. Experimental results against $\epsilon$=4.0 are left to the Appendix.

## 5. Experimental Setting

We apply 10-step PGD (Madry et al., 2018) to attack the defenses with $\epsilon = 4.0/255$ and 1.0/255. The backbone model is ViT-B/32 for all papers except CLIPure (Zhang et al., 2025), which uses ViT-L/14 in the main experiments and TAPT (Wang et al., 2025), which uses ViT-B/16 in the original paper. To test the robustness, we randomly choose 1000 data points from each of 13 datasets following FARE (Schlarmann et al., 2024) and CLIPure (Zhang et al., 2025). Note that some methods report the results on different splits of the dataset, thus there can be some fluctuations on the absolute values of the results. However, this difference doesn't affect

the generalizability of the relative values. All of the experiments are based on public code of the defenses. We trained the adversarial model for alignment following instruction of TAPT and we conduct the experiments on one A6000 GPU.

## 6. Extra Experimental Analysis

**Visualization**    We provide visualizations of the indicative measurement $m(\mathbf{x})$ for TTC and R-TPT in Figure 3. On TTC, $m(\mathbf{x})$ for CLIP-MAD "clean" data imitates the distribution of adversarial data to induce unexpected feature modification. On R-TPT, $m(\mathbf{x})$ for CLIP-MAD adversarial data imitates the distribution of clean data for unexpected feature preservation. The visualizations in Figure 3(a) and Figure 3(b) correspond to two ways of achieving our goal of indicative measurement manipulation to fool the defense.

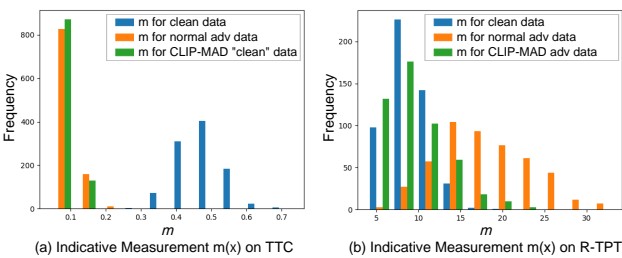

(a) Indicative Measurement $m(\mathbf{x})$ on TTC     (b) Indicative Measurement $m(\mathbf{x})$ on R-TPT

*Figure 3.* Distribution of adversarial indicative measurement for clean, normal adversarial and CLIP-MAD testing samples on TTC and R-TPT defenses

**More Advanced Attack Baselines**    We provide additional results of our CLIP-MAD against more complex attack baselines other than PGD-10. Table 6 shows the classification accuracy against APGD (one component of AutoAttack)

*Table 6.* Zero-shot robust accuracies under APGD Restart=2 attack with $\epsilon = 4.0/255$ for defense method R-TPT

| | CIFAR10 | CIFAR100 | STL10 | StanfordCars | PCAM | Food101 | OxfordPet | Flowers102 | Caltech101 | Caltech256 | EuroSAT | TinyImageNet | ImageNet |
|---|---|---|---|---|---|---|---|---|---|---|---|---|---|
| Static | 34.2 | 17.6 | 75.8 | 11.8 | 31.2 | 46.2 | 45.8 | 41.2 | 74.6 | 58.8 | 2.2 | 11.8 | 37.6 |
| **CLIP-MAD (Ours)** | **0.2** | **0.0** | **2.4** | **0.0** | **2.4** | **0.0** | **0.2** | **1.4** | **2.2** | **14.0** | **0.0** | **0.0** | **0.6** |

*Table 7.* Zero-shot robust accuracies under PGD Restart=3 EOT-10 attack with $\epsilon = 4.0/255$ for defense method TTC

| | CIFAR10 | CIFAR100 | STL10 | PCAM | OxfordPet | Caltech101 | Flowers102 | Food101 | EuroSAT | TinyImageNet | StanfordCars |
|---|---|---|---|---|---|---|---|---|---|---|---|
| Static | 29.8 | 9.6 | 56.9 | 51.1 | 24.4 | 61.3 | 14.9 | 18.5 | 5.9 | 5.9 | 9.1 |
| Full | 12.0 | 2.7 | 26.8 | 27.7 | 8.7 | 25.7 | 5.2 | 3.8 | 2.2 | 1.0 | 3.8 |
| **CLIP-MAD (Ours)** | **3.9** | **0.0** | **10.6** | **20.5** | **2.3** | **13.2** | **0.8** | **0.1** | **0.0** | **0.0** | **0.3** |

Restart=2 $\epsilon = 4.0/255$ attack on R-TPT defense (on the first half of the testing data in previous section). Table 7 shows classification accuracy (on the same testing data in previous section) against PGD Restart=3 EOT-10 $\epsilon = 4.0/255$ attack on TTC. Both results show the efficacy of our CLIP-MAD. The reason is our CLIP-MAD finds an attack direction that remains unexpected and less affected by the defense on more advanced and computationally intensive adversarial optimization baselines.

**Attack Strength against Standard Preprocessing** To demonstrate the robustness of our CLIP-MAD attack against standard preprocessing techniques, we provide results against JPEG compression on R-TPT defense in Table 9 and results against geometric transformation on TTC defense in Table 10 in Appendix A.2. Note that in the original implementation of CLIP-MAD for TTC, we deliberately choose a milder version of $\mathcal{L}_m$ that implicitly modifies $m(\mathbf{x})$ to make the least unnecessary perturbation. To preserve attack strength against random transformation, we make a slight modification of $\mathcal{L}_m$ that includes the term for explicitly reducing $m(\mathbf{x})$. We didn't provide JPEG compression experiment on TTC because JPEG compression is detrimental to TTC. (JPEG compression of quality 85 already reduces robust accuracy from 24.3% to 11.5% with $\epsilon = 4.0/255$ full gradient attack on STL10.) The results show that the proposed CLIP-MAD remains effective against general preprocessing techniques.

**Limitations and Practical Values** Similar to other adaptive attack methods, CLIP-MAD requires more knowledge of the test-time defense than that required by grey-box or black-box attacks, which may restrict its convenience in applications. Despite this limitation, there are certain aspects of CLIP-MAD that make it valuable in real scenarios. First, our CLIP-MAD can be used as a tool for future designers of test-time defenses to evaluate the inner vulnerability of the defense. If the defense can not bear this simple method of

crafting adaptive attacks and the indicative measurement is easily manipulated, the defense can be far from the essential traits of adversarial samples. Second, our CLIP-MAD lowers the barriers of conducting adaptive attacks. Compared to full gradient attacks that require full information of the defense including complete optimization details, our CLIP-MAD skips defense operations once the indicative measurement is exposed or recognized. This gives our method the capability of constructing efficient adaptive attacks even if the following defensive steps are hard to compute or inaccessible.

## 7. Discussion and Conclusion

Our findings on the fragile *Indicative-Measurement Assumptions* in CLIP's test-time defenses indicate three shifts for future research:

- **Avoiding Unreliable Indicators:** Dramatic performance degradation against our CLIP-MAD would imply unreliable adversarial indicative measurement.

- **Conducting Diversified Evaluation:** Instead of always trying to attack the whole defense, researchers may consider the shortcut of attacking the indicative measuring step, especially when full or approximated gradient attack fails.

- **Utilizing Generalization for Defense:** Test-time defenders may consider aggressive and hard-to-approximate preprocessing. The generalization abilities of large-scale VLM makes this possible without severely harming performance.

Ultimately, our work finds a shared but previously overlooked vulnerability of zero-shot test-time defenses and provides an efficient tool to check for this potential weakness. We hope our work can help future researchers reach the goal of truly resilient zero-shot inference.

## Acknowledgement

This work was supported by the Institute Innovation Project of the Institute of Computing Technology, Chinese Academy of Sciences under Grant No. E561090.

## Impact Statement

Our work presents an vulnerability of current state-of-the-art test-time defenses for CLIP. In real world foundation model applications, this vulnerability may be maliciously utilized by attackers once the basic assumption of the defense is known by the attackers. Our CLIP-MAD provides researchers an efficient tool to check for this potential weakness during defense design and can be very useful for safety and goodness of the society.

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

# A. Appendix

## A.1. Additional Experimental Results

In Table 8, we present the experimental results on defenses CLIPure-Cos and CLIPure-Diff with $\epsilon$=4.0/255.

*Table 8.* Zero-shot robust accuracies under PGD-10 attack with $\epsilon$=4.0/255 for defense methods CLIPure-Cos and CLIPure-diff.

| Defense | Attack | CIFAR10 | CIFAR100 | STL10 | Food101 | Oxford | Flowers102 | EuroSAT | TinyImageNet | ImageNet | Caltech101 | Caltech256 | StanfordCars | PCAM | Avg |
|---|---|---|---|---|---|---|---|---|---|---|---|---|---|---|---|
| | Full | 86.6 | 64.2 | 98.4 | 88.8 | 80.1 | 74.7 | 35.7 | 54.9 | 87.2 | 99.0 | 94.8 | 56.1 | 55.4 | 75.1 |
| CLIPure-Cos | Static | 0.3 | 0.0 | **2.6** | 0.0 | 0.0 | 0.0 | 0.0 | 0.0 | 0.0 | 0.0 | 4.7 | 0.0 | 0.2 | 0.6 |
| | **Ours** | **0.3** | **0.0** | 2.7 | **0.0** | **0.0** | **0.0** | **0.0** | **0.0** | **0.0** | **0.1** | **4.3** | **0.0** | **0.2** | **0.6** |
| | Full | 92.7 | 72.0 | 96.1 | 93.6 | 76.2 | 81.7 | 88.0 | 65.2 | 86.9 | 85.9 | 95.4 | 54.7 | 58.8 | 80.1 |
| CLIPure-Diff | Static | 1.1 | 0.1 | 3.9 | 0.0 | 0.0 | 1.0 | 0.0 | 0.0 | 0.0 | 0.4 | 4.4 | 0.0 | 0.1 | 0.9 |
| | **Ours** | **1.1** | **0.0** | **3.3** | **0.0** | **0.0** | **0.0** | **0.0** | **0.0** | **0.0** | **0.0** | **4.2** | **0.0** | **0.0** | **0.7** |

## A.2. Attack against Standard Preprocessing

Table 9 and Table 10 present the robust accuracies of the zero-shot test-time defenses enhanced by input preprocessing defensive techniques prior to inference. Our CLIP-MAD attack remains effective against this combination of defenses.

*Table 9.* Robust accuracies of JPEG compression (quality parameter=85) and R-TPT against adversarial attacks (first half of testing data)

| | STL10 | Caltech101 | Caltech256 | PCAM | OxfordPet | Food101 |
|---|---|---|---|---|---|---|
| Static | 77.4 | 77.0 | 60.2 | 40.4 | 45.4 | 48.8 |
| **CLIP-MAD (Ours)** | **7.0** | **4.4** | **16.6** | **2.0** | **1.2** | **0.4** |

*Table 10.* Robust accuracies of input transformation (RandomResizedCrop(scale=(0.5,1.0)) and RandomHorizontalFlip) and TTC against adversarial attacks

| | STL10 | PCAM | Caltech101 | Caltech256 | OxfordPet | Food101 | tinyImageNet | ImageNet |
|---|---|---|---|---|---|---|---|---|
| Static | 56.3 | 18.0 | 56.2 | 41.5 | 19.9 | 22.8 | 3.4 | 20.1 |
| Full | 41.0 | 16.0 | 51.0 | 34.2 | 17.2 | 17.6 | 1.9 | 21.2 |
| **CLIP-MAD (Ours)** | **15.8** | **12.6** | **18.6** | **9.9** | **3.1** | **1.6** | **0.4** | **6.2** |

## A.3. List of Datasets

The experimental results are reported on 13 datasets: CIFAR10 (Krizhevsky & Hinton, 2009), CIFAR100 (Krizhevsky & Hinton, 2009), STL10 (Coates et al., 2011), TinyImagenet (Deng et al., 2009), ImageNet (Deng et al., 2009), Caltech101 (Li et al., 2004), Caltech256 (Griffin et al., 2007), StanfordCars (Krause et al., 2013), PCAM (Bejnordi et al., 2017), Food101 (Bossard et al., 2014), OxfordPets (Parkhi et al., 2012), Flowers102 (Nilsback & Zisserman, 2008) and EuroSAT (Helber et al., 2019).

## A.4. Hyperparameters

The hyperparameters $\alpha$ and $\lambda$ can be set easily without complex hyperparameter tuning. During our experiments, $\lambda$ is naturally set to 1 as $\alpha$ is 0 for TTC and DOC. When $\alpha$ is set to 1, $\lambda$ is also set to 1 for equal emphasis on different views for TAPT, R-TPT, CLIPure-Cos and CLIPure-Diff. Our CLIP-MAD is effective even if $\alpha$ (the hyperparameter for basic static loss) is set to 0 for defenses like TTC. The reason is TTC and DOC follow the paradigm of recognizing adversarial samples and devastating the "adversarial" features (via counter-attack perturbation). Thus in the scenario of adaptive attack, once CLIP-MAD manipulates the indicative measurement, the defense produce unexpected behavior of attacking benign features by itself. The stronger the counterattack, the lower the performance. This corresponds to one way of applying our CLIP-MAD: manipulating "clean" features' indicative measurement to simulate adversarial features for unexpected results.

Note that this technique targets the test-time defended model instead of static model (pretrained model without defense) as we are constructing adaptive attacks.

