# OpenReview forum: "You Don't Protect if You Don't Expect: Breaking the Key Assumption behind CLIP's Test-Time Defenses"
_ICML.cc/2026/Conference — ICML 2026 regular_

### Official Review · Reviewer_xadD · 2026-03-06

**Soundness:** 3
**Presentation:** 3
**Significance:** 3
**Originality:** 3
**Overall Recommendation:** 4
**Confidence:** 3

**Summary:**

This paper demonstrates that the adversarial robustness of current CLIP test-time defenses is significantly overestimated due to a shared reliance on fragile indicative metrics for threat detection. To expose this critical flaw, the authors introduce CLIP-MAD, an adaptive attack utilizing a dual-objective loss to manipulate these metrics and shatter the defenses' distributional assumptions. By circumventing end-to-end gradient computation, CLIP-MAD easily bypasses non-differentiable or computationally heavy defense components. Empirical results show that the proposed method reliably defeats mainstream defenses—even when traditional full-gradient attacks fail—thereby exposing the false sense of security in existing pipelines and providing a standardized benchmark for evaluating zero-shot CLIP robustness.

**Compliance With Llm Reviewing Policy:**

Affirmed.

**Final Justification:**

The additional experiment and justificaition furtuer supported the proposed method, which raise my confidence of this method.

**Key Questions For Authors:**

1. Practical Threat Model and Black-Box Viability
While white-box access is valuable for establishing theoretical upper bounds on attack efficacy, it rarely reflects practical deployment scenarios, which are predominantly gray- or black-box. What specific real-world threat model does this paper aim to address? Furthermore, in restricted settings where internal states are opaque, how would the proposed method accurately estimate or obtain the target system's indicator metric $m(x)$ to maintain its attack viability?
2. Robustness Against Standard Preprocessing
Do the adversarial examples generated against these specific test-time defenses exhibit cross-defense robustness when subjected to simple, fixed-preprocessing defense mechanisms prior to inference (e.g., uniform JPEG compression or standard geometric transformations)?
3. Insufficient Evidence for Attack Orthogonality
The manuscript claims that CLIP-MAD is orthogonal to the underlying adversarial attack method; however, the empirical evaluation relies exclusively on PGD-10. This single baseline is insufficient to substantiate the broad claim of orthogonality. If confronted with a highly robust defender necessitating more advanced, computationally intensive attack algorithms, would the proposed framework still maintain its claimed efficiency and effectiveness?

**Limitations:**

yes

**Strengths And Weaknesses:**

1. Soundness
Strengths: Technically rigorous and heavily supported by extensive evaluations (6 SOTA defenses across 13 datasets). Effectively bypasses gradient obfuscation and non-differentiable components where traditional attacks fail. Authors demonstrate objective self-assessment.
Limitations: Lacks comprehensive attack baselines, critical hyperparameter ablations, physical-world evaluations, and detailed failure-case analyses.

2. Presentation
Strengths: Well-structured, highly reproducible, and clearly written. Related works are well-positioned, and experimental visual aids are cleanly integrated.
Limitations: Sparse appendix, minor typographical errors in formulas, and insufficient in-depth discussion of classic foundational literature.

3. Significance and Originality
Strengths: Systematically exposes a critical, shared vulnerability across three major categories of CLIP test-time defenses. The highly generalizable and efficient CLIP-MAD framework establishes a crucial new baseline for zero-shot robustness evaluation.
Limitations: Restricted to white-box settings, lacks proposed defensive countermeasures, and the core adaptive attack logic relies somewhat incrementally on prior studies.

---

> ### Author Rebuttal · Authors · 2026-03-31
>
> Thanks for your comments. Here is our response:
> 1. **Practical Usage**: We agree with your opinion that in the real case, most attacks are conducted in the grey-box or black-box way and adaptive attacks are valuable for establishing theoretical upper bounds on attack efficacy. However, we still hope you may consider the following scenarios that our proposed CLIP-MAD can be useful: Our CLIP-MAD can be used as a tool for future researches of test-time defenses to expose the inner vulnerability of the defense. As we mentioned in the paper, if the defense can not bear this simple strategy of crafting adaptive attacks and the indicative measurement is easily manipulated, the defense can be far from the essential traits of adversarial samples.
>
> 2. **Robustness against Standard Preprocessing**: We provide results against JPEG compression on R-TPT and results against geometric transformation on TTC:
> - Results against JPEG compression of quality parameter=85 on R-TPT (first half of testing data).
>
> |          |STL10|Caltech101|Caltech256|PCAM|OxfordPet|Food101
> |:-----:|:-----:|:-----:|:-----:|:-----:|:-----:|:-----:|
> |Static  |  77.4   |    77.0    |60.2|40.4|45.4|48.8|
> |CLIP-MAD  |   **7.0**   |    **4.4**| **16.6**    |**2.0**|**1.2**|**0.4**|
>
> - Results against RandomResizedCrop(scale=(0.5,1.0)) and RandomHorizontalFlip on TTC.
> In the original implementation of CLIP-MAD for TTC, we deliberately chose a milder version of $\mathcal{l}_m$ that implicitly modifies m(x) to make the least unnecessary perturbation. To preserve attack strength against random transformation, we make a slight modification of $\mathcal{l}_m$ that includes the term of explicitly reducing m(x). Here are the results:
>
> |        |STL10|PCAM| Caltech101|Caltech256|OxfordPet|Food101|tinyImageNet|ImageNet
> |:-----:|:-----:|:-----:|:-----:|:-----:|:-----:|:-----:|:-----:|:-----:|
> |Static|56.3|18.0|56.2| 41.5|19.9|22.8|3.4|20.1|
> |Full|41.0|16.0| 51.0|34.2|17.2|17.6|1.9|21.2|
> |CLIP-MAD|**15.8**|**12.6**|**18.6**| **9.9** |**3.1**|**1.6**|**0.4**|**6.2**|
>
> We didn’t conduct JPEG compression on TTC because JPEG compression is detrimental to TTC defense. (JPEG compression of quality 85 already reduces robust accuracy from 24.3% to 11.5% with $\epsilon$=4.0/255 full gradient attack on STL10.)
>
> 3. **More Advanced Attack Baseline**: We provide additional results against more complex attack baselines other than PGD-10:
>
> - The table below shows classification accuracy against APGD Restart=2 $\epsilon$=4.0/255 attack on R-TPT:
>
> |          | CIFAR10 | CIFAR100 | STL10 | StanfordCars | PCAM | Food101 | OxfordPet | Flowers102 |Caltech101  | Caltech256  |EuroSAT  | tinyImageNet |  ImageNet|
> |:-------|:-----:|:-----:|:-----:|:-----:|:-----:|:-----:|:-----:|:-----:|:-----:|:-----:|:-----:|:-----:|:-----:|
> | Static   | 34.2      | 17.6       | 75.8    | 11.8          | 31.2  | 46.2      |45.8        | 41.2     |74.6  |      58.8  |     2.2  |     11.8  |      37.6|
> | CLIP-MAD | **0.2**     |**0.0**      | **2.4**     |**0.0**  | **2.4**  | **0.0**     | **0.2**         |  **1.4**  | **2.2**   |     **14.0** |       **0.0**   |    **0.0** |        **0.6**|
>
> Due to the time constraint, the results in the table above is on the first half of the testing data in the paper.
>
> - The table below shows classification accuracy against PGD Restart=3 EOT-10 $\epsilon$=4.0/255 attack on TTC:
>
> |          | CIFAR10| CIFAR100| STL10| PCAM| OxfordPet| Caltech101| Flowers102|Food101|EuroSAT|tinyImageNet| StanfordCars|
> |:-------|:-----:|:-----:|:-----:|:-----:|:-----:|:-----:|:-----:|:-----:|:-----:|:-----:|:-----:|
> | Static| 29.8|  9.6| 56.9| 51.1| 24.4| 61.3| 14.9| 18.5|5.9|5.9|9.1|
> | Full| 12.0 | 2.7| 26.8 | 27.7|  8.7 |  25.7| 5.2|3.8|2.2|1.0|3.8|
> | CLIP-MAD| **3.9** | **0.0**  | **10.6** | **20.5** |  **2.3** | **13.2** | **0.8** | **0.1** |**0.0**  |**0.0** | **0.3** |
>
> The results in the table above is on the same testing data of the paper.
> These two tables shows that our CLIP-MAD are effective across more advanced and computational intensive attack baselines.

---

> > ### Author Rebuttal · Reviewer_xadD · 2026-04-01
> >
> > My concerns have been adequately addressed.

---

### Official Review · Reviewer_6VpQ · 2026-03-07

**Soundness:** 3
**Presentation:** 3
**Significance:** 4
**Originality:** 3
**Overall Recommendation:** 4
**Confidence:** 2

**Summary:**

This paper investigates the robustness of six recently proposed test-time defenses for CLIP's zero-shot classification. The authors argue that the claimed robustness of these methods is significantly overestimated because they share a fundamental vulnerability: a reliance on an "indicative measurement" to distinguish between clean and adversarial samples. To exploit this, the authors propose CLIP-MAD, an adaptive attack strategy designed to directly manipulate this assumed measurement, thereby inducing the defense to misclassify or fail to purify the input. The paper evaluates this attack across 13 datasets, demonstrating severe drops in the reported robust accuracy of the targeted defenses.

**Compliance With Llm Reviewing Policy:**

Affirmed.

**Key Questions For Authors:**

1.  Why was the empirical evaluation restricted to PGD-10 instead of utilizing comprehensive, standard attack suites like AutoAttack or multi-restart PGD to truly establish worst-case robustness, is it due to they are hard to perform under this paper's setting?
2.  For defenses that rely on stochastic augmentations (such as R-TPT), why was Expectation over Transformation (EoT) omitted from the baseline adaptive attacks? Can you provide EoT-based results in the rebuttal?
3.  Can you provide empirical visualizations (e.g., histograms) of the measurement $m(x)$ distributions for clean vs. adversarial samples to substantiate the conceptual claims made in Equations (6)-(14)?
4.  When setting $\alpha=0$ in the CLIP-MAD objective (as done for TTC), how does the attack inherently guarantee that the resulting sample will be misclassified by the static model, rather than just successfully evading the purification gate?

**Limitations:**

Yes, the authors briefly discuss the potential negative societal impact of exposing vulnerabilities in foundation models in the Impact Statement.

**Strengths And Weaknesses:**

**Strengths:**

* **Significance & Originality:** The paper makes a highly valuable conceptual contribution by abstracting a common denominator—the "indicative measurement" assumption—across highly diverse test-time defense paradigms (input purification, feature optimization, prompt tuning). This unified perspective provides a strong foundation for understanding why current defenses fail.
* **Breadth of Target Datasets:** The evaluation across 13 distinct image classification datasets provides a wide and convincing lens for assessing zero-shot robustness vulnerabilities.

**Weaknesses (Methodological and Evaluation Gaps):**

* **Limited Attack Baselines (Soundness & Significance):** While the conceptual framework is strong, a significant limitation of the current evaluation is its restricted scope. The authors rely entirely on PGD-10 ($\epsilon=1.0/255$ and $\epsilon=4.0/255$). For evaluating "worst-case" adversarial robustness, relying solely on a low-iteration PGD attack is inadequate. The paper would be substantially stronger with evaluations against standard suites like AutoAttack, CW, or multi-restart PGD. There is also a lack of comparisons with other advanced adaptive attack strategies beyond "Static" and "Full-gradient".
* **Treatment of Stochastic Defenses (Soundness):** Several targeted defenses, such as R-TPT, utilize stochastic operations like AugMix. To rigorously evaluate stochastic defenses, adaptive attackers typically employ Expectation over Transformation (EoT). The authors bypass this by substituting differentiable proxies during attack generation. Without EoT-based evaluations, the robustness degradation might be partially attributed to the proxy substitution rather than a complete break of the defense mechanism.
* **Theoretical Claims Lack Empirical Verification (Soundness):** The paper uses a clean mathematical framework to describe how distributions of the indicative measurement, $p_{\phi_{clean}}(m(x))$ and $q_{\phi_{adv}}(m(x))$, differ. However, this framework (Equations 6-14) remains largely conceptual. The paper misses the opportunity to provide empirical data—such as density plots or histograms of $m(x)$ before and after the attack—to visually and quantitatively substantiate the claim that CLIP-MAD successfully forces adversarial samples into the assumed clean distribution.
* **Ambiguity in Loss Formulation:** For certain defenses like TTC, the authors set the classification loss weight $\alpha$ to $0$, focusing solely on manipulating the measurement $m(x)$. The paper needs to better justify how shifting $m(x)$ strictly guarantees a misclassification event, rather than simply bypassing the defense's gating mechanism to yield a benign static feature.

---

> ### Author Rebuttal · Authors · 2026-03-31
>
> Thanks for your positive comments. Here is our explanation for your questions:
> 1. **Limited Attack Baselines**:
>
> We have added results of APGD (one component of AutoAttack) Restart=2 attacks on R-TPT and PGD Restart=3 EOT-10 attacks on TTC to demonstrate the effectiveness of our CLIP-MAD across attack baselines. The table below shows classification accuracy against attacks based on APGD Restart=2 $\epsilon$=4.0/255 on R-TPT:
>
> |          | CIFAR10 | CIFAR100 | STL10 | StanfordCars | PCAM | Food101 | OxfordPet | Flowers102 |Caltech101  | Caltech256  |EuroSAT  | tinyImageNet |  ImageNet
> |:-------|:-----:|:-----:|:-----:|:-----:|:-----:|:-----:|:-----:|:-----:|:-----:|:-----:|:-----:|:-----:|:-----:|
> | Static   | 34.2      | 17.6       | 75.8    | 11.8  | 31.2  | 46.2      |45.8 | 41.2     |74.6  |      58.8  |     2.2  |     11.8  |      37.6|
> | CLIP-MAD | **0.2**     | **0.0**      |**2.4**     |**0.0** | **2.4** | **0.0** | **0.2** |  **1.4** | **2.2**   | **14.0** |  **0.0**   | **0.0** | **0.6**|
>
> Due to the time constraint, the results in Table 1 is on the first half of the testing data in the paper. However, this doesn't affect our conclusion.
> In the following table, we provide classification accuracy against attacks based on PGD Restart=3 EOT-10 $\epsilon$=4.0/255 on TTC:
> |          | CIFAR10| CIFAR100| STL10| PCAM| OxfordPet| Caltech101| Flowers102|Food101|EuroSAT|tinyImageNet| StanfordCars|
> |:-------|:-----:|:-----:|:-----:|:-----:|:-----:|:-----:|:-----:|:-----:|:-----:|:-----:|:-----:|
> | Static| 29.8|  9.6| 56.9| 51.1| 24.4| 61.3| 14.9| 18.5|5.9|5.9|9.1|
> | Full| 12.0 | 2.7| 26.8 | 27.7|  8.7 |  25.7| 5.2|3.8|2.2|1.0|3.8|
> | CLIP-MAD| **3.9** | **0.0**  | **10.6** | **20.5** |  **2.3** | **13.2** | **0.8** | **0.1** |**0.0**  |**0.0** | **0.3** |
>
> The results in Table 2 is on the same testing data of the paper. Both tables support the effectiveness of method across different baselines.
>
> 2. **Expectation over Transformation**:
> TTC, DOC, R-TPT and TAPT contain stochasticity through random noise or augmentations. The original TTC and DOC Full and CLIP-MAD results reported in the paper is already integrated with EOT-5. We have added EOT-10 results on TTC in Table 2 provided above. We’d like to explain why EoT is not integrated with R-TPT and TAPT. Basically, R-TPT and TAPT don’t generate a single random augmentation but a batch containing 64 augmentations for each input. Then the prediction is averaged and tuned on the average of confident ones from the 64 random augmentations. Thus to conduct full defense gradient attack, one need to generate and average predictions from a batch of random augmentations in each attack iteration, which has similar effects with EoT that takes the average from multiple random results. However, this process is cumbersome for both the augmentation and later index selection based reweighting creates non-differentiability or discontinuity. Our CLIP-MAD is flexible in this case for it skips defense operations once the indicative measurement is recognized. This gives our method the capability to construct efficient attacks even if the following defense computation is hard to compute. In the case of R-TPT, CLIP-MAD can be regarded as a simplified EoT that reduces the difference against augmentation between clean and adversarial samples.
>
> 3. **Empirical Verification**:
>  We provide visualizations of the indicative measurement m(x) for TTC and R-TPT to show that the results support the conceptual claims.
> - Distribution of indicative measurement $m(\mathbf{x})$ on TTC:  https://imgur.com/a/EMmIivz
> - Distribution of indicative measurement $m(\mathbf{x})$ on R-TPT:  https://imgur.com/a/6W1ygV5
>
> As an explanation for the visualization, on TTC, $m(\mathbf{x})$ for CLIP-MAD “clean” data imitates the distribution of adversarial data to induce unexpected feature modification. On R-TPT, $m(\mathbf{x})$ for CLIP-MAD adversarial data imitates the distribution of clean data for unexpected feature preservation. The visualizations show two ways of achieving our goal of indicative measurement manipulation to fool the defense.
>
> 4. **Ambiguity of Loss Formulation**:
> We would like to explain why is our method effective even if α is set to 0 for TTC. The reason is TTC and DOC follow the paradigm of recognizing adversarial samples and devastating the adversarial features (via self-supervised counter-attack perturbation). Thus in the scenario of adaptive attack, once CLIP-MAD manipulates the indicative measurement, the defense produce unexpected behavior of attacking benign features by itself. The stronger the counterattack perturbation, the lower the performance. This corresponds to one way of applying our CLIP-MAD: manipulating “clean” features’ indicative measurement to simulate adversarial features for unexpected results. Note that this technique targets the test-time defended model instead of static model (pretrained model without defense) as we are constructing adaptive attacks.

---

> > ### Author Rebuttal · Reviewer_6VpQ · 2026-04-01
> >
> > My concerns have been adequately addressed.

---

### Official Review · Reviewer_PsBh · 2026-03-08

**Soundness:** 3
**Presentation:** 3
**Significance:** 4
**Originality:** 2
**Overall Recommendation:** 4
**Confidence:** 4

**Summary:**

The paper evaluates the robustness of 6 defenses against adaptive attacks on the CLIP model.
Specifically, the authors develop a modular attack strategy called CLIP-MAD that can break the defenses simultaneously.
The authors also run extensive experiments across 13 datasets showing that the attack can effectively and consistently reduce the robust accuracy of the defended CLIP model.
Overall, the paper shows that current SOTA defenses do not sufficiently protect CLIP from adversarial attacks.

**Compliance With Llm Reviewing Policy:**

Affirmed.

**Final Justification:**

I have a positive view of the paper, and the authors have taken the time to address all concerns I had. There are some weaknesses of the paper (e.g., methodological novelty) but in my opinion these weaknesses are overwhelmingly outweighed by the practical impact of the findings and the adaptability of the attack, which form the core strengths of the paper.

**Key Questions For Authors:**

1. Could the authors clarify what the limitations of their method are?

**Limitations:**

As far as I can tell, the authors have not discussed any limitations of their method or work.
I would appreciate if the authors could do so in their rebuttal.

**Strengths And Weaknesses:**

This was a very pleasant paper to read and very easy to follow.
The results are also extensive and show a substantial vulnerability in the defenses studied and the methodology is elegant and adaptable.
The only major weakness I would say of the paper is that their method amounts to modifying the loss function used to find adversarial samples to take into account the defenses, a common strategy used in prior work on adversarial attacks.
Nevertheless, I believe the impact of the findings and the adaptability of the attack outweighs the deficiency in methodological novelty.

I have a couple of suggestions for the authors to improve their paper, mainly on the presentation side.
1. Line 37, I believe the sentence "its application to CLIP introduces significant drawbacks..." requires a citation.
2. Line 177, The wrong inverted commas have been used for "adversarial trait".
3. The term "robust accuracy" is never defined properly.

---

> ### Author Rebuttal · Authors · 2026-03-31
>
> Thanks for your positive comments.
> - **citation**: We will add citation of the following paper for this sentence: Clip is strong enough to fight back: Test-time counterattacks towards zero-shot adversarial robustness of clip. CVPR 2025. This paper explains why adversarial finetuning should by employed sparingly for large pretrained models.
> - **grammar mistakes**: We will revise the comma mistake in our paper.
> - **undefined terms**: The term “robust accuracy” in our work means “classification accuracy on adversarial images” in previous works.  We have added this to next version of our paper.
> - **limitations**: Similar to other adaptive attack methods, our CLIP-MAD requires certain knowledge of the test-time defense. Specifically, our method need the indicative measurement utilized by the defense. Despite this limitation, our work can effectively enhance the strength of adaptive attacks in zero-shot classification once the indicative measurement of the defense is known.

---

> > ### Author Rebuttal · Reviewer_PsBh · 2026-04-01
> >
> > thank you to the authors for resolving my concerns! I do not think any of my initial concerns were particularly major so I will be keeping my score.

---

### Official Review · Reviewer_RmwM · 2026-03-10

**Soundness:** 4
**Presentation:** 4
**Significance:** 3
**Originality:** 3
**Overall Recommendation:** 4
**Confidence:** 4

**Summary:**

This paper has a key insight that many recently proposed test-time defenses for CLIP have a key assumption that there is some distributional difference across clean and adversarial examples with regards to some measurement.  The authors propose CLIP-MAD which integrates this measurement into the optimization objective for the attack, thus making the examples no longer abide by this assumption.  They demonstrate that this insight can be applied across 5 different SOTA defenses and is effective across a large number of datasets.

**Compliance With Llm Reviewing Policy:**

Affirmed.

**Key Questions For Authors:**

N/A paper is quite clear

**Limitations:**

Yes

**Strengths And Weaknesses:**

Strengths:
- writing is clear and organized
- authors demonstrate how their insight can be applied to 5 different test time attacks and experimentally validate that their attacks can reduce the robust accuracy of these defenses across different datasets

Weaknesses:
- limited citations: I think one important citation that is missing is: Carlini and Wagner 2017 "Adversarial examples are not easily detected: Bypassing ten detection methods".  I think the insight and attack technique is a bit similar to their insight and attack for the defense of using a KDE detector to detect adversarial examples (they include a penalty for being detected into the objective, which I think is similar to minimizing an observed behavior discrepancy between the 2 distributions).
- significance feels a bit limited since the core takeaway of needing to evaluate against adaptive attacks is already established in this field, this work mainly demonstrates this for CLIP's test time defenses.

---

> ### Author Rebuttal · Authors · 2026-03-31
>
> Thanks for your positive comments. Here is our response:
> 1. **limited citations**: Thanks for pointing out this related work. We will add the reference of the paper you mentioned to our Related Works section in the following way: Previous work [1] in traditional adversarial robustness provides valuable insights that the intrinsic property of adversarial samples is hard to capture and many existing defenses can be bypassed by carefully constructed adversarial samples, which inspires us to explore the resilience of the newly proposed paradigm of test-time defenses in zero-shot classification.
> 2. **significance of method**: Although previous works in traditional classification have pointed out that adaptive attacks can reduce the performance of test-time defenses, there exist some differences between previous works and ours:
> - The setting of zero-shot robustness poses new challenges and potential vulnerability to test-time defenses. As far as we know, our work is the first to analyze adaptive attacks in zero-shot classification.
> - Our work further enhances the strength of adaptive attacks and proposes a simple formulation that can be easily adapted to different defenses.

---

> > ### Author Rebuttal · Reviewer_RmwM · 2026-04-01
> >
> > I didn't have any major concerns so I will keep my score

---

### Decision · Program_Chairs · 2026-04-30

**Decision:**

Accept (regular)

**Comment:**

The authors evaluate the robustness of 6 defenses against adaptive attacks on the CLIP model. Specifically, this paper develops a modular attack strategy called CLIP-MAD that can break the defenses simultaneously. After the discussion phase, all the concerns are addressed by the authors, ensuring the contributions made by this paper to the field.